# Constrained Reweighting of Distributions: An Optimal Transport Approach

**DOI:** 10.3390/e26030249

**Published:** 2024-03-11

**Authors:** Abhisek Chakraborty, Anirban Bhattacharya, Debdeep Pati

**Affiliations:** Department of Statistics, Texas A&M University, College Station, TX 77843, USA; anirbanb@stat.tamu.edu (A.B.); debdeep@stat.tamu.edu (D.P.)

**Keywords:** complex surveys, demographic parity, entropy, optimal transport, portfolio allocation

## Abstract

We commonly encounter the problem of identifying an optimally weight-adjusted version of the empirical distribution of observed data, adhering to predefined constraints on the weights. Such constraints often manifest as restrictions on the moments, tail behavior, shapes, number of modes, etc., of the resulting weight-adjusted empirical distribution. In this article, we substantially enhance the flexibility of such a methodology by introducing a nonparametrically imbued distributional constraint on the weights and developing a general framework leveraging the maximum entropy principle and tools from optimal transport. The key idea is to ensure that the maximum entropy weight-adjusted empirical distribution of the observed data is close to a pre-specified probability distribution in terms of the optimal transport metric, while allowing for subtle departures. The proposed scheme for the re-weighting of observations subject to constraints is reminiscent of the empirical likelihood and related ideas, but offers greater flexibility in applications where parametric distribution-guided constraints arise naturally. The versatility of the proposed framework is demonstrated in the context of three disparate applications where data re-weighting is warranted to satisfy side constraints on the optimization problem at the heart of the statistical task—namely, portfolio allocation, semi-parametric inference for complex surveys, and ensuring algorithmic fairness in machine learning algorithms.

## 1. Introduction

The maximum entropy principle [1,2] states that in situations characterized by uncertainty and limited prior knowledge-guided constraints, the optimal choice among all feasible probability distributions is the probability distribution that is the least informative or most uniformly spread. This idea is at the heart of numerous statistical tasks and has permeated into every corner of modern machine learning research. Prominent instances of such constrained entropy maximization include applications in image reconstruction [3], ill-posed inverse problems [4], portfolio optimization [5], generalized methods of moment models [6], natural language processing [7], network analysis [8], and reinforcement learning [9], to name a few. We refer the readers to Cover and Thomas [10], Kardar [11] for book-length reviews.

For maximum entropy inference, the specified constraints imposed on the probability distributions frequently manifest as constraints pertaining to moments [6], tail characteristics [12], distributional shapes [13], modal counts, and similar properties. In many cases, however, constructing constraints with the desired level of flexibility is challenging, if not unfeasible—refer to Section 3 and Section 5 for specific examples in the context of inference in complex surveys and moment conditions based on portfolio optimization, respectively. On a related note, a recent article [14] introduced a flexible framework for the introduction of more elaborate constraints on probability distributions in the context of conducting robust Bayesian inference.

In this article, we offer a novel solution to this problem via introducing a probability distribution-guided constrained entropy maximization framework that not only offers versatility but also enhances the interpretability of the inferential output. The main concept revolves around ensuring that a weight-adjusted empirical distribution of the observed data closely aligns with a predetermined family of probability distributions, measured through a statistical distance [15]. Importantly, the family of probability distributions is potentially continuous, but any weighted-adjusted empirical distribution of the observed data is discrete. This eliminates the possibility of adopting many common statistical discrepancies, e.g., Kullback–Leibler, total variation, or Hellinger’s distance, to place the probability distribution-guided constraints. In practice, we need to exercise extreme care to ensure that our choice is tailored to the application of interest. For homogeneity of exposition across all scenarios in this article, we consider the Wasserstein metric [16,17].

The idea of data re-weighting is, of course, not new. Ref. [18] suggested elevating the likelihood of individual observations using data-driven weights, to conduct robust inference under mild model misspecification. Ref. [19] proposed a data re-weighting scheme to align the data with a different target distribution, enabling inference under covariate shifts. Other compelling ideas involving re-weighting can be found in fair learning [20], natural language processing [21], variational tempering [22], etc. Complementing the existing literature, we propose a versatile data re-weighting framework, borrowing from the maximum entropy principle and optimal transport, that is useful in a multitude of statistical tasks.

The rest of the paper is organized as follows. The general framework of the proposed probability distribution-guided constrained entropy maximization is introduced in Section 2. Section 3, Section 4 and Section 5 present applications of our methodology in the context of semi-parametric inference in complex surveys, in ensuring demographic parity in machine learning algorithms, and entropy-based portfolio optimization, respectively. Finally, we conclude with a discussion.

## 2. General Framework

Let [a] denote the set of integers {1,…,a}. Let Ω denote the set of all possible discrete distributions ω with atoms $s={\left(s_{1},\ldots ,s_{m}\right)}^{T}$. The entropy of the discrete probability distribution $\sum_{i=1}^{m}w_{i}\delta_{s_{i}}\left(\cdot \right)$ is defined by
$$\begin{array}{c} H_{m}\left(w\right)=-\sum_{i=1}^{m}w_{i}\log w_{i}, \end{array}$$
where δ is Dirac’s delta function. The entropy $H_{m}\left(w\right)$ is a measure of randomness, which is maximized at the discrete uniform distribution with $w_{i}=1/m$ for all *i*. In many statistical tasks, the core challenge constitutes optimizing a functional $F:\Omega \to \Omega^{\prime }$ with respect to ω subject to a constraint $\omega \in \Omega_{0}\left(\subset \Omega \right)$. A simple example is when $s={\left(s_{1},\ldots ,s_{m}\right)}^{T}$ is the observed sample itself. Then, the set Ω is simply characterized by the class of weighted empirical distributions of the observed data,
$$\begin{array}{c} \Omega =\left\{\omega =\sum_{j=1}^{m}w_{j}\;\delta_{s_{j}}\left(\cdot \right):\sum_{j=1}^{m}w_{j}=1,\;w_{j}\geq 0,\;j\in \left[m\right]\right\}. \end{array}$$
In the following, we shall provide more general examples where the constraint set $\Omega_{0}$ can be identified with a subset of an (m−1)-dimensional probability simplex $S_{m-1}=\left\{w:\sum_{i=1}^{m}w_{i}=1,\;w_{i}>0,\;i\in \left[m\right]\right\}$, for some m∈{1,2,…}.

Given $s={\left(s_{1},\ldots ,s_{n}\right)}^{T}$, parametric inference constitutes approximating the empirical distribution $\left(1/n\right)\sum_{i=1}^{n}\delta_{s_{i}}\left(\cdot \right)$ via a parametric family of distributions $\left\{f_{\theta }:\theta \in \Theta \right\}$, and learning the parameter θ from data. Such procedures often fall prey to model misspecification [23], leading to untrustworthy inferences. To avoid complete model specification, a popular class of semi-parametric approaches [24] operate under a milder assumption that the weight-adjusted empirical distribution $\sum_{i=1}^{n}w_{i}\delta_{s_{i}}\left(\cdot \right)$ satisfies moment restrictions of the form $\sum_{i=1}^{n}w_{i}\;g\left(s_{i},\theta \right)=0$, where *g* is a vector of known functions on $R^{d}\times \Theta$. In numerous instances, achieving such moment-based constraints with the intended degree of flexibility proves to be arduous, if not practically impossible—we elaborate on this more in the sequel. To that end, in this article, we offer a middle ground between the fully parametric and semi-parametric moment condition models that allows for flexible modeling assumptions while enjoying coherent interpretability similar to parametric inference. We propose to operate under a restriction of the form $D(\sum_{i=1}^{n}w_{i}\delta_{s_{i}}\left(\cdot \right),f_{\theta })\leq \varepsilon$, where D is a statistical distance and ε is a user-defined hyperparameter. Our goal is to infer θ while allowing for mild deviations from the parametric model $f_{\theta }$, and ε measures the maximum allowable discrepancy.

Inference under moment condition models often involves computing the maximum entropy weight-adjusted empirical distribution of ${\left(s_{1},\ldots ,s_{n}\right)}^{T}$ that satisfies some pre-specified moment conditions [6,25]. That is, for every θ∈Θ, we calculate $\sum_{i=1}^{n}w_{i}^{★}\left(\theta \right)\;\delta_{s_{i}}\left(\cdot \right)$, where $w^{★}\left(\theta \right)={arg max}_{w\in S_{n-1}}H_{n}\left(w\right)$, subject to $\sum_{i=1}^{n}w_{i}g\left(s_{i},\theta \right)=0$. Under the proposed framework, we too appeal to the maximum entropy principle and compute the maximum entropy weight-adjusted empirical distribution of s that satisfies the parametric distribution-guided constraint. That is, for every θ∈Θ, we calculate $\sum_{i=1}^{n}w_{i}^{★}\left(\theta \right)\;\delta_{s_{i}}\left(\cdot \right)$, where
(1)$$\begin{array}{cc} \; & w^{★}\left(\theta \right)=\underset{w\in S_{n-1}}{arg max}H_{n}\left(w\right)\;\mathrm{subject}\;\mathrm{to}\;D\left(f_{\theta },\;\sum_{i=1}^{n}w_{i}\;\delta_{s_{i}}\left(\cdot \right)\right)\leq \varepsilon , \end{array}$$
where D is a statistical distance, and ε is a user-defined parameter. In the ensuing applications in this article, we often solve the dual optimization problem for operational ease. In this case, for each θ∈Θ and λ≥0, we calculate $\sum_{i=1}^{n}w_{i}^{★}\left(\theta \right)\;\delta_{s_{i}}\left(\cdot \right)$ such that
(2)$$\begin{array}{c} w^{★}\left(\theta \right)=\underset{w\in S_{n-1}}{arg max}\left[H_{n}\left(w\right)-\lambda \;D\left(f_{\theta },\;\sum_{i=1}^{n}w_{i}\;\delta_{s_{i}}\left(\cdot \right)\right)\right]. \end{array}$$
The parameter λ controls the extent of departure from the guiding parametric distribution.

One pivotal aspect yet to be addressed within the proposed framework is that a weight-adjusted empirical distribution is discrete, but, in the context of a specific problem, the guiding distribution $f_{\theta }$ is potentially continuous. For instance, in Section 5, in the context of entropy-based portfolio allocation, $f_{\theta }$ takes the form of a skew normal distribution [26]. This precludes the utilization of several standard statistical distances, such as total variation, Hellinger’s distance, $\chi^{2}$ distance, etc., for the implementation of the distance-based constraint. In this article, due to its versatility, we employ the Wasserstein metric [17] with the $L_{2}$ cost as the distance measure D. To that end, we briefly recall some relevant facts about the 2-Wasserstein metric. The Wasserstein space $P_{2}\left(R^{d}\right)$ is defined as the set of probability measures μ with finite moment of order 2, i.e., $\left\{\mu \;:\;\int_{R^{d}}{∥x∥}^{2}d\mu \left(x\right)<\infty \right\}$, where $∥\;\cdot \;∥$ is the Euclidean norm on $R^{d}$.

**Definition** **1**.*For $p_{0},p_{1}\in P_{2}\left(R^{d}\right)$, let $\pi \left(p_{0},p_{1}\right)\subset P_{2}\left(R^{d}\times R^{d}\right)$ denote the subset of joint probability measures (or *couplings*) ν on $R^{d}\times R^{d}$ with marginal distributions $p_{0}$ and $p_{1}$, respectively. Then, the 2-Wasserstein distance $W_{2}$ between $p_{0}$ and $p_{1}$ is defined as $W_{2}^{2}\left(p_{0},p_{1}\right)=\inf_{\nu \in \pi (p_{0},p_{1})}\int_{R^{d}\times R^{d}}\;{∥y_{0}-y_{1}∥}^{2}\;d\nu \left(y_{0},y_{1}\right)$.*

Importantly, if both $p_{0},p_{1}\in P_{2}\left(R\right)$ with quantile functions $F_{0}^{-1},F_{1}^{-1}$, we have a highly tractable expression [27]: $W_{2}^{2}\left(p_{0},p_{1}\right)=\int_{\left[0,1\right]}{\left[F_{0}^{-1}\left(q\right)-F_{0}^{-1}\left(q\right)\right]}^{2}dq.$ This is heavily utilized in the subsequent sections. With this, we have all the essential components to delve into the specific applications of interest.

The central idea of re-weighting observations subject to constraints is reminiscent of the empirical likelihood framework (EL; [28,29,30,31]) for conducting non-parametric inference. An EL approximates the underlying distribution with a discrete distribution supported at the observed data points and obtains the induced maximum likelihood of the parameter of interest defined through constraints, by effectively profiling out the nuisance parameters. Qin and Lawless [32] hugely expanded the scope of EL by integrating it with estimating equations. EL has also been adapted to the dependent data setup; see Nordman and Lahiri [33] for a detailed review in the context of time series data. Operationally, EL-based methods enjoy great computational simplicity. The computation of the EL at θ∈Θ amounts to solving the convex optimization problem $\{\max_{w}\sum_{i=1}^{n}\log w_{i}|w_{i}>0,\;\sum_{i=1}^{n}w_{i}=1,$$\sum_{i=1}^{n}w_{i}g\left(s_{i},\theta \right)=0\}$. On the theoretical side, [34] demonstrated that the EL estimator exhibits desirable higher-order asymptotic properties in a well-specified setup. Newey and Smith [34] also showed that the EL estimator may not be $\sqrt{n}$-convergent when g(·) is unbounded. On the other hand, Schennach [35] showed that the exponentially tilted empirical likelihood (ETEL) attains the same asymptotic bias and variance as EL, as well as retaining $\sqrt{n}$-convergence under model misspecification. ETEL minimizes the Kullback–Leibler divergence of this discrete distribution with the empirical distribution of the observed data subject to satisfying the estimating equation. Operationally, ETEL-based methods continue to enjoy the same computational simplicity as EL, since its computation at θ∈Θ amounts to solving the convex optimization problem $\{\max_{w}\sum_{i=1}^{n}-w_{i}\log w_{i}|w_{i}>0,$$\sum_{i=1}^{n}w_{i}=1,\;\sum_{i=1}^{n}w_{i}g\left(s_{i},\theta \right)=0\}$.

The direct application of the ETEL optimization routine to our setup is challenging as the moment conditions describing the parameters of general parametric models, especially those beyond exponential families, can be quite cumbersome or even unavailable. Instead, our approach proceeds by constraining a weighted empirical distribution of the observed data $\sum_{i=1}^{n}w_{i}\delta_{s_{i}}$ to be close to the parametric model $F_{\theta }$ with respect to a statistical metric. The key advantage of the proposed framework lies in the greater flexibility that it offers compared to existing EL or ETEL procedures based on moment conditions. Let us explain this with the application of the proposed framework to the portfolio optimization problem. Volatility Feedback Theory [36] posits that market volatility can influence subsequent returns. Specifically, it suggests that periods of high volatility can lead to skewed returns, where extreme price movements are more likely to occur. This theory underscores the idea that heightened volatility tends to be associated with increased risk and uncertainty in financial markets, potentially resulting in non-normal return distributions characterized by asymmetry and fat tails. Skewed returns indicate that the probability of extreme events, either positive and negative, is higher than what would be expected under a normal distribution. Consequently, skew normal distributions are routinely utilized to model observed returns [37,38]. This is an example where a distributional constraint arises naturally, explaining the added utility of the proposed framework.

The added flexibility potentially comes at a cost. The constraint of the form $D(f_{\theta },\;\sum_{i=1}^{n}$$w_{i}\;\delta_{s_{i}}\left(\cdot \right))\leq \varepsilon$ is non-linear in the weights, posing a greater computational challenge in computing the ETEL. However, we find that augmented Lagrangian methods [39,40] and conic solvers [41] via the R interface [42] of constrained non-linear optimization solvers (for example, NLopt (Johnson [43]) and CVX (Grant and Boyd [44])) serve our purpose in the class of problems that we consider.

On the theoretical front, it is indeed interesting to study the asymptotic properties of the ETEL estimator under our distributional constraint and compare it to the asymptotic properties of the ETEL estimators with moment constraints that were eluded to earlier. Such study, however, will come with unique challenges due to the potentially non-convex nature of the set $\left\{{\left(w_{1},\ldots ,w_{m}\right)}^{T}:D\left(f_{\theta },\;\sum_{i=1}^{n}w_{i}\;\delta_{s_{i}}\left(\cdot \right)\right)\leq \varepsilon ,\sum_{j=1}^{m}w_{j}=1,\;w_{j}\geq 0,\;j\in \left[m\right]\right\}$. In particular, the asymptotic bias and higher-order variance of the ETEL estimator subject to moment-based constraints are obtained via obtaining the stochastic expansions of the ETEL estimator of the parameter of interest and the Lagrange multiplier [34]. These stochastic expansions are in turn obtained by first studying the consistency and asymptotic normality of the estimator as prerequisites. To develop similar large-sample properties for the ETEL estimator subject to distributional constraints, the existing techniques are not directly useful. We need to devise appropriate regularity conditions to develop the required stochastic expansions from scratch. Thus, it is well beyond the scope of the current article and presents an opportunity for future investigations.

An instance of application of the proposed framework emerges within the realm of semi-parametric inference in complex survey data [45,46]. In survey sampling, we wish to infer about a collection of features of a finite population $P:=\{X_{i},\;i\in \left[N\right]\}$. We are provided with a non-representative sample $\left(x_{1},\ldots ,x_{n}\right)$ obtained from P via a complex survey, and the corresponding survey weights $\pi =\left(\pi_{1},\ldots ,\pi_{n}\right),\;0<\pi_{i}<\infty$. In the general framework, this task involves finding the optimal $\omega \in \Omega_{0}\subset \Omega$ such that
$$\begin{array}{c} \Omega =\left\{\omega =\sum_{j=1}^{n}w_{j}\delta_{s_{i}}\left(\cdot \right):\sum_{j=1}^{n}w_{j}=1,w_{j}\geq 0,\;j\in \left[n\right]\right\}, \end{array}$$
where $\left(s_{1},\ldots ,s_{n}\right)=\left(x_{1}^{★},\ldots ,x_{n}^{★}\right)$ is an i.i.d. pseudo-sample of size *n* obtained from the complex survey sample $\left(x_{1},\ldots ,x_{n}\right)$, via weighted finite population Bayesian bootstrap [47,48,49] to adjust for the survey weights, and m=n. The restriction $\Omega_{0}$ is dictated by the parametric model that the analyst posits on finite population P to infer about the features of interest in the finite population.

The next application in this article deals with the issue of ensuring demographic parity [50,51] in machine learning algorithms. Suppose that we have data $\left(x_{i},y_{i},a_{i}\right)\in X\times Y\times \left\{S,T\right\}$ for *n* individuals on covariate $x\in R^{p}$, continuous response y∈R, and *protected/sensitive* attribute *A* with labels {S,T}. For the sake of simplicity in exposition, we further assume that $a_{i}=S,\;i\in \left[n_{S}\right]$, $a_{i}=T,\;i\in \left[n\right]\setminus \left[n_{T}\right]$ and $n=n_{S}+n_{T}$. The goal is to learn a predictive rule, h:X×{S,T}→Y, that satisfies the specific notion of demographic parity. Refer to Section 4 for details. We shall see that this task involves finding the optimal $\omega \in \Omega_{0}\subset \Omega$ such that
$$\begin{array}{c} \Omega =\left\{\omega =\sum_{j=1}^{n_{T}}w_{n_{S}+j}\;\delta_{s_{j}}\left(\cdot \right):\sum_{j=1}^{n_{T}}w_{n_{S}+j}=1,w_{n_{S}+j}\geq 0,\;j\in \left[n_{T}\right]\right\}, \end{array}$$
where $s_{j}=-L\left(\theta_{\left(T\right)}|x_{j}\right),\;j\in \left[n\right]\setminus \left[n_{S}\right]$ is the negative of the loss function utilized to learn the predictive rule *h* for individuals with a=T, and $\theta_{\left(T\right)}$ are the associated parameters. The optimality of ω and restriction $\Omega_{0}$ are determined by the notion of demographic parity utilized.

An application of a slightly modified version of the general framework is identified in portfolio allocation problems [5,52,53], where the goal is to identify the optimal atoms of the discrete distributions, rather than the weights assigned to the atoms. This task translates to finding the optimal $\omega \in \Omega_{0}\subset \Omega$ such that
$$\begin{array}{c} \Omega =\left\{\omega =\frac{1}{n}\sum_{i=1}^{n}\delta_{s_{i}}\left(\cdot \right):\sum_{j=1}^{d}w_{j}=1,w_{j}\geq 0,\;j\in \left[d\right]\right\}, \end{array}$$
where $s_{i}=\sum_{j=1}^{d}w_{j}r_{i,j},\;i\in \left[n\right]$; refer to Section 5 for details. The optimality criterion and the restriction $\Omega_{0}$ are driven by the fund manager’s portfolio allocation objectives and the assumed model for the return distribution, respectively.

## 3. Semi-Parametric Inference in Complex Surveys

Survey data [45,46] commonly arise from complex sampling methods such as stratification and multistage sampling, wherein individuals in the finite population have unequal probabilities of inclusion in the sample. Prominent instances of extensive surveys implementing these methodologies include the National Health and Nutrition Examination Surveys (NHANES), the British Household Panel Survey (BHPS), the Household Income and Labour Dynamics in Australia (HILDA) survey, etc. In complex surveys, the survey sample lacks representativeness, since the individuals with varying demographic characteristics in the finite population of interest have varying probabilities of selection in the sample. Consequently, traditional methods of inference and estimation result in bias and the poor coverage of estimators.

A prevalent approach to addressing this challenge entails carefully exploiting the sampling weights available with complex survey datasets. These weights could be used to rectify the biases introduced by the unequal probability sampling and enable us to create pseudo-equal probability samples from the population. If a survey participant falls within a demographic group with a low probability of selection or response, their weight is increased accordingly. Commonly, the available information only includes the survey dataset and the associated sampling weights for each unit in the sample. That is, there is limited or no information available about the complex sampling methodology or the precise technique employed to derive these weights. This situation presents a compelling inferential challenge, which we shall delve into further in the following discussion.

Assume that we have a finite population $P:=\{X_{i},\;i\in \left[N\right]\}$, and we wish to infer about a collection of features of P. We are provided with a non-representative sample $x=(x_{1},\ldots ,x_{n})$ obtained via a complex survey design, and the corresponding survey weights $\pi =(\pi_{1},\ldots ,\pi_{n}),$
$0<\pi_{i}<\infty$. It is assumed that the weights have been designed so that $\pi_{i}$ is inversely proportional to the likelihood that the survey design selects an observation with the same demographic characteristics as observation $x_{i}$. That is, observations with a lower probability of being selected than they would have under a simple random sampling approach are assigned greater weight than they would receive in a simple random sampling scenario. Conversely, observations with a higher probability of selection receive lower weights than they would in a simple random sampling setup. The $\pi_{i}$s are scaled to ensure that $\sum_{i=1}^{n}\pi_{i}=n$.

### 3.1. Related Works

Pseudo-maximum likelihood (PMLE)-based approaches are very popular with regard to conducting parametric inference with complex survey data, where we posit a parametric model $f_{\theta }$ to model P and θ encodes the features of interest of P. The pseudo-loglikelihood of θ takes the form $L\left(\theta \right)=\sum_{i=1}^{n}\pi_{i}\log f_{\theta }\left(x_{i}\right)$ [45,54]. The pseudo-likelihood estimate of ${\hat{\theta }}_{\mathrm{PMLE}}$ satisfies the first-order condition $\frac{\partial L(\theta )}{\partial \theta }=\sum_{i=1}^{n}\pi_{i}\frac{\partial }{\partial \theta }\log f_{\theta }\left(x_{i}\right)$. Under a certain regularity condition [23],
$$\begin{array}{c} \sqrt{n}\left({\hat{\theta }}_{\mathrm{PMLE}}-\theta_{0}\right)\overset{d}{\to }N\left(0,H_{\pi }^{-1}V_{\pi }H_{\pi }^{-1}\right), \end{array}$$
where $\theta_{0}$ is the true value of θ, and $H_{\pi }$ and $V_{\pi }$ are estimated by
$$\begin{array}{cc} \; & {\hat{H}}_{\pi }=\frac{1}{n}\sum_{i=1}^{n}\pi_{i}\frac{\partial^{2}}{\partial \theta \partial \theta^{T}}\log f_{\theta }\left(x_{i}\right){|}_{\theta ={\hat{\theta }}_{\mathrm{PMLE}}},\\ \; & {\hat{V}}_{\pi }=\frac{1}{n}\sum_{i=1}^{n}\pi_{i}\frac{\partial \log f_{\theta }\left(x_{i}\right)}{\partial \theta }\frac{\partial \log f_{\theta }\left(x_{i}\right)}{\partial \theta^{T}}{|}_{\theta ={\hat{\theta }}_{\mathrm{PMLE}}} \end{array}$$
respectively.

As an alternative, a semi-parametric inference framework can be developed where the feature of interest θ of the finite population $P:=\{X_{i},\;i\in \left[N\right]\}$, instead of a parametric family of distributions, as earlier, is described by the set of estimating equations $\frac{1}{N}\sum_{i=1}^{N}g\left(X_{i},\theta \right)=0$, with a vector of known functions *g*. This approach avoids the complete parametric specification of the model, and it is widely utilized in statistics and econometrics [6,25]. Given a sample $x={\left(x_{1},\ldots ,x_{n}\right)}^{T}$ and survey weights $x={\left(\pi_{1},\ldots ,\pi_{n}\right)}^{T}$, the exponentially tilted empirical likelihood [55] is given by
$$\begin{array}{c} L_{\mathrm{MCM}}\left(\theta \right)=\left\{\prod_{i=1}^{n}w_{i}^{★}:w^{★}=\underset{w}{arg max}H_{n}\left(w\right),\;w\in S_{n-1},\;\sum_{i=1}^{n}w_{i}\left[\pi_{i}g\left(x_{i},\theta \right)\right]=0\right\}. \end{array}$$
Here, and elsewhere, we use MCM as an acronym for the *moment condition model*. When the convex hull of $\cup_{i=1}^{n}g\left(x_{i},\theta \right)$ contains the origin, it leads to $L_{\mathrm{MCM}}\left(\theta \right)=\prod_{i=1}^{n}w_{i}^{★}\left(\theta \right)$, with
$$\begin{array}{c} w_{i}^{★}\left(\theta \right)=\frac{\exp[\pi_{i}\lambda {\left(\theta \right)}^{T}g\left(x_{i},\theta \right)]}{\sum_{j=1}^{n}\exp\left[\pi_{j}\lambda {\left(\theta \right)}^{T}g\left(x_{j},\theta \right)\right]} \end{array}$$
and $\lambda \left(\theta \right)={arg min}_{\eta }n^{-1}\sum_{i=1}^{n}\exp\left[\pi_{i}\eta^{T}g\left(x_{i},\theta \right)\right]$. When the convex hull condition is not satisfied, $L_{\mathrm{MCM}}\left(\theta |x_{1},\ldots ,x_{n}\right)$ is set to zero.

### 3.2. Proposed Methodology

Importantly, it is often difficult, if not impossible, to place more flexible constraints on the parameter of interest via moment conditions. In this article, we intend to provide additional flexibility to the ETEL framework via providing the scope for statistical distance-based parametric distribution-guided constraints. However, it is not straightforward to accomplish this in the context of complex survey data, due to the presence of the survey weights. To carefully circumnavigate this issue, we first reconstruct *M* pseudo-true populations of size *N* from the observed complex survey sample of size *n* via weighted finite population Bayesian bootstraping [47,48,49] to adjust for the survey weights; next, we draw an i.i.d. pseudo-sample of size *n* from each of the pseudo-true populations, and we finally construct an ETEL based on each of the *M* pseudo-samples. Given the *m*-th i.i.d. pseudo-sample $\left(x_{m,1}^{★},\ldots ,x_{m,n}^{★}\right),\;m\in \left[M\right]$, the exponentially tilted empirical likelihood with a parametric distribution-guided constraint takes the form
$$\begin{array}{cc} L_{\mathrm{BDCM}}\left(\theta \right)= & \{\prod_{i=1}^{n}w_{i}^{★}:w^{★}=\underset{w}{arg max}H_{n}\left(w\right),\;w\in S_{n-1},\\ \; & \sum_{i=1}^{n}w_{i}g\left(x_{m,i}^{★},\theta \right)=0,\;W_{2}^{2}\left[\sum_{i=1}^{n}w_{i}\delta_{x_{i}^{★}}\left(\cdot \right),\;f_{\theta }\right]\leq \varepsilon \}, \end{array}$$
where δ is the indicator function, $f_{\theta }$ is the parametric distribution of choice, and ε is a user-defined parameter denoting the maximum extent of departure from the parametric distribution of choice. Here, and elsewhere, we use BDCM as an acronym for *bootstrapped distributionally constrained models*. Importantly, the inference on the *M* pseudo-true samples can be carried out in parallel. The final estimate of θ is obtained via combining the estimates obtained from the *M* i.i.d. pseudo-samples.

### 3.3. Experiments

Based on the numerical experiments in [45], we design simulation studies to compare the proposed distribution-guided entropy maximization approach with the popular pseudo-likelihood approach. Suppose that the random variables (X,Z) jointly follow a bivariate normal distribution with mean ${\left(\mu_{x},\mu_{z}\right)}^{\prime }={\left(0,10\right)}^{\prime }$, marginal variances ${\left(\sigma_{x}^{2},\sigma_{z}^{2}\right)}^{\prime }={\left(4,16\right)}^{\prime }$, and correlation ρ∈{0.1,0.5,0.8}. The variable *X* is the variable of interest; we aim to estimate its mean $\mu_{x}$ and variance $\sigma_{x}^{2}$. The variable *Z* is a selection variable, i.e., the *Z*-value of a population unit determines the probability of inclusion of the unit in the sample. Specifically, we posit that the inclusion probability of $X_{s}$ in the sample is given by $\pi_{s}^{★}=\Phi \left(\beta_{0}+\beta_{1}Z_{s}\right)$, where Φ(·) is the cumulative distribution function of a standard normal distribution. When a population unit is included in the sample, we observe $x_{s}$ and assign a survey weight $\pi_{s}$ such that $\pi_{s}\propto 1/\pi_{s}^{★}$. Importantly, we assume that we do not directly observe $Z_{s}$. The selected sample of size *n* is denoted as ${\left(x,\pi \right)}^{\prime }$. We scale the weights such that they sum up to 1, and we have $\pi_{s}=\frac{\left(1/\pi_{s}^{★}\right)}{\sum_{j=1}^{n}\left(1/\pi_{j}^{★}\right)},\;s\in \left[n\right]$. The objective is to utilize ${\left(x,\pi \right)}^{\prime }$ to estimate the population parameters of interest $\left(\mu_{x},\sigma_{x}^{2}\right)$.

We generate N=100,000 values of $\left(X_{s},Z_{s}\right)$ as a finite population. We set $\beta_{0}=0.1,$
$\beta_{1}=-1.8$ and draw samples of sizes n∈{500,1000,1500,2000,2500} from the finite population. Under each data generating setup, we utilize 100 Monte Carlo simulations. For the pseudo-maximum likelihood (PMLE) approach, we simply posit the model $f_{\theta }\equiv \mathrm{Normal}\left(\mu_{x},\sigma_{x}^{2}\right)$. For the proposed BCDM approach, we assume the moment constraint based on the function $g\left(x,\mu_{x}\right)=x-\mu_{x}$, and the Wasserstein distance constraint based on the parametric family of distributions $f_{\theta }\equiv \mathrm{Normal}\left(\mu_{x},\sigma_{x}^{2}\right)$. For each of the replicates, we choose M=500; to ensure the comparability of PMLE and BDCM, we set $\varepsilon =W_{2}^{2}\left[\sum_{i=1}^{n}1/n\delta_{x_{i}^{★}}\left(\cdot \right),\;f_{\hat{\theta }}\right]$, where $\hat{\theta }$ is the estimate of θ obtained via PMLE. The bias and the coverage of the pseudo-maximum likelihood and moment condition model-based estimators for varying data generating mechanisms are presented in Table 1. A case study with complex survey data from the National Health and Nutrition Examination Surveys (NHANES) is provided later.

### 3.4. National Health and Nutrition Examination Surveys (NHANES) Data Analysis

The NHANES is a series of surveys designed to assess the health and nutritional status of individuals in the United States. The data extracted are from the NHANES 2009–2010 [46], and they contain information on binary indicators of high cholesterol, race, age, etc., and survey weights for 8591 individuals. For this exercise, we assume that these 8591 individuals make up a finite population and obtain samples of size n∈{250,500,1000,2000} according to the survey weights. We fit a logistic regression to model the binary indicator of high cholesterol as a function of race and age. For each n∈{250,500,1000,2000}, we utilize 100 Monte Carlo simulations. For the distribution-guided entropy maximization approach, we assume constraints on the score function of the logistic regression. The coverage of the moment condition model-based estimates of the regression coefficients is presented in Table 2.

## 4. Demographic Parity

Discrimination pertains to the unfair treatment of individuals based on specific demographic characteristics known as protected attributes. The goal of demographic parity or statistical parity [50,51] in machine learning is to design algorithms that yield fair inferences devoid of discrimination due to membership in certain demographic groups determined by a protected attribute. First, we introduce the mathematical formalization of the notions of demographic parity. To that end, we assume that *X* denotes the feature vector used for predictions, *A* is the protected attribute with two levels {S,T}, and *Y* is the response. Parity constraints are phrased in terms of the distribution over (X,A,Y). Two definitions are in order.

**Definition** **2**(Demographic parity, [50]). *A predictor h satisfies demographic parity under the distribution over (X,A,Y) if h(X) is independent of the protected attribute A, i.e., P[h(X)≥z∣A=S]=P[h(X)≥z∣A=T]=P[h(X)≥z], for all z.*

**Definition** **3**(Demographic parity in expectation, [50]). *A predictor h satisfies demographic parity under the distribution over (X,A,Y) if h(X) is independent of the protected attribute A, i.e., E[h(X)∣A=S]=E[h(X)∣A=T]=E[h(X)].*

### 4.1. Proposed Methodology

Although the notions of demographic parity in Definitions 2 and 3 coincide when we work with binary responses, the latter may be amenable to simple computational algorithms [57] compared to the general definition. However, the notion of demographic parity in expectation is somewhat prohibitive since one cannot control the predictor *h* over its entire domain. For example, depending on the application of interest, we may be solely interested in controlling the tails of the predictor [58]. Returning to our semi-parametric inference framework, we offer a flexible as well as a computationally feasible compromise between the notions in Definitions 2 and 3. To that end, we introduce the notion of demographic parity in the Wasserstein metric next.

**Definition** **4** (Demographic parity in Wasserstein metric).
*A predictor h achieves demographic parity in the Wasserstein metric with bias ε, under the distribution over (X,A,Y), if $W_{2}^{2}\left[F_{h_{S}},F_{h_{T}}\right]\leq \varepsilon$, where $F_{h_{k}}$ is the empirical distribution of h under sub-population k, i.e., h(X)∣A=k,k∈{S,T}.*


Suppose that we have data $\left(x_{i},y_{i},a_{i}\right)\in R^{d}\times R\times \left\{S,T\right\}$ for *n* individuals on *p*-dimensional covariate *x*, univariate continuous response *y*, and the levels of the protected attribute a∈{S,T}. For the sake of simplicity in exposition, we also assume that $a_{i}=S,\;i\in \left[n_{S}\right]$ and $a_{i}=T,\;i\in \left[n\right]\setminus \left[n_{S}\right]$, where $n=n_{S}+n_{T}$. Next, we posit a predictive model, $y_{i}=h\left(x_{i},\theta_{\left(a_{i}\right)}\right)+e_{i},\;e_{i}\overset{i.i.d}{∼}N\left(0,\sigma^{2}\right),\;i\in \left[n\right]$, where *h* is potentially non-linear and $\left(\theta_{\left(S\right)},\theta_{\left(T\right)}\right)$ is the model parameter of interest to be estimated under the demographic parity constraint $W_{2}^{2}\left[F_{h_{S}},F_{h_{T}}\right]\leq \varepsilon$. In particular, we consider the empirical CDF of *h* under sub-population *S*, $F_{h_{S}}=1/n_{S}\sum_{i=1}^{n_{S}}\delta_{h(x_{iS})}\left(\cdot \right)$, and a weighted empirical CDF of *h* under sub-population *T*, $F_{h_{T}}=\sum_{i=n_{S}+1}^{n}w_{i}\;\delta_{h(x_{iT})}\left(\cdot \right)$. Here, δ is the Dirac delta measure. The goal is to infer about $\left(\theta_{\left(S\right)},\theta_{\left(T\right)},w\right)$, ensuring that the demographic parity constraint, i.e., $F_{h_{S}},F_{h_{T}}$, is close with respect to $W_{2}^{2}$, and, at the same time, the extent of re-weighting in $F_{h_{T}}$ is minimal, i.e., the entropy $-\sum_{i=n_{S}+1}^{n}w_{i}\log w_{i}$ is close to the maximal entropy $\log n_{T}$. A related idea in Jiang et al. [59] deals with $W_{1}$ constrained fair classification problems, but our approach of additionally re-weighting the observations offers more flexibility, with possible ramifications in the study of fairness in misspecified models.

We achieve this through an *in-model* approach solving the optimization problem
(3)$$\begin{array}{c} \max_{w,\theta_{\left(S\right)},\theta_{\left(T\right)},\sigma^{2}}[-\frac{1}{n_{S}}\sum_{i=1}^{n_{S}}l_{i}\left(\theta_{\left(S\right)}|x_{i}\right)-\sum_{i=n_{s}+1}^{n}w_{i}l_{i}\left(\theta_{\left(T\right)}|x_{i}\right)\\ -\left(1-\lambda^{★}\right)W_{2}^{2}\left[F_{h_{S}},F_{h_{T}}\right]-\lambda^{★}\sum_{i=n_{s}+1}^{n}w_{i}\log w_{i}] \end{array}$$
where $\sum_{i=n_{s}+1}^{n}w_{i}=1$ and $l_{i}\left(\theta_{\left(a_{i}\right)}|x_{i}\right)={\left(y_{i}-h\left(x_{i},\theta_{\left(a_{i}\right)}\right)\right)}^{2}/2\sigma^{2},\;i\in \left[n\right]$. For a resulting re-weighting vector $w^{★}={\left(w_{n_{S}+1}^{★},\ldots ,w_{n}^{★}\right)}^{\prime }$, we can obtain a fair prediction at a new x∈T via a weighted kernel density estimate at *x*. As a competitor to the *in-model* scheme, motivated by popular post-processing schemes to ensure fairness [60,61], we utilize a *two-step* procedure.

**Step 1:** We obtain model parameter estimates by $({\hat{\theta }}_{\left(S\right)},\;{\hat{\theta }}_{\left(T\right)},\;{\hat{\sigma }}^{2})=$
(4)$$\underset{\theta_{\left(S\right)},\;\theta_{\left(T\right)},\;\sigma^{2}}{arg max}\left[-\frac{1}{n_{S}}\sum_{i=1}^{n_{S}}l_{i}\left(\theta_{\left(S\right)}|x_{i}\right)-\frac{1}{n_{T}}\sum_{i=n_{s}+1}^{n}l_{i}\left(\theta_{\left(T\right)}|x_{i}\right)\right]$$
followed by a post-processing step at $\left({\hat{\theta }}_{\left(S\right)},\;{\hat{\theta }}_{\left(T\right)},\;{\hat{\sigma }}^{2}\right)$ to obtain $w^{★}$


**Step 2:**

(5)
$$\underset{w}{arg max}\left[-\left(1-\lambda^{★}\right)\;W_{2}^{2}\left[F_{h_{S}},F_{h_{T}}\right]-\lambda^{★}\sum_{i=n_{s}+1}^{n}w_{i}\log w_{i}\right].$$



A case study on algorithmic mental health monitoring is provided next. An additional case study on algorithmic criminal risk assessment is also included.

### 4.2. Distress Analysis Interview Corpus (DAIC)

The Distress Analysis Interview Corpus (DAIC) [62] is a multi-modal clinical interview collection, accessible upon request via the DAIC-WOZ (https://dcapswoz.ict.usc.edu/ (accessed on 16 October 2023)) website. Computer agents based on such clinical interviews are deemed to be used to make mental health diagnoses in relation to certain employment decisions, and concerns about the fairness of such tools with respect to the biological gender of the individuals have been raised. Specifically, we focus on predicting the PHQ-8 score, which captures the individual’s severity of depression, as a function of the individual’s verbal signals during the clinical interviews, while biological gender serves as a protected attribute. In particular, the Fourier series analysis of the speech signals of the individuals yields verbal attributes of interest, which in turn could be potentially used in the diagnosis of the individual’s severity of depression. Therefore, it is of interest to develop novel methods to produce predictions while avoiding disparate treatment on the basis of biological gender. More precisely, we wish to ensure that the demographic parity constraint is satisfied here, which, in this context, simply dictates that the weighted empirical CDFs of the biological gender-specific fitted PHQ-8 scores are *identical* or *similar*.

The PHQ-8 scores range from 0 to 27, with a score of 0–4 considered none or minimal, 5–9 mild, 10–14 moderate, 15–19 moderately severe, and 20–27 severe. In this application, we work with the PHQ-8 (continuous response), biological gender (binary protected attribute), and 17 derived audio/verbal features (continuous covariates) corresponding to the n=107 subjects. The PHQ-8 scores for the two biological genders show a clear discrepancy. Therefore, we shall assess the relative performance of the *in-model* scheme (Equation 3) and the *two-step* scheme (Equation 4) and (Equation 5) in ensuring demographic parity with respect to biological gender (refer to Figure 1 and Figure 2. As earlier, for the sake of simplicity of exposition, we use linear regression (i.e., *h* is linear in the covariates) as our predictive model of choice. When we fit the predictive model without any fairness constraint, the fitted empirical cumulative distribution functions corresponding to the two biological genders are widely different. Our *in-model* scheme, as well as the *two-step*, scheme significantly reduces the discrepancy owing to its in-built fairness-based regularization. As noted earlier, the *in-model* scheme provides lower bias since it performs the two-step optimization simultaneously.

### 4.3. COMPAS Recidivism Data Analysis

We consider a case study of algorithmic criminal risk assessment. We shall focus on the popular COMPAS dataset [63], which includes information on the criminal history of defendants in Broward County, Florida, available from the propublica website (https://www.propublica.org/datastore/dataset/compas-recidivism-risk-score-data-and-analysis (accessed on 16 October 2023)). For each individual, several features of the criminal history are available, such as the number of past felonies, misdemeanors, and juvenile offenses; additional demographic information includes the sex, age, and ethnic group of each defendant. We focus on predicting the two-year recidivism score *y* (continuous) as a function of the defendant’s demographic information, except for race and criminal history *x*, while race (categorical) serves as a protected attribute. Algorithms for the creation of such predictions are routinely used in courtrooms to advise judges, and concerns about the fairness of such tools with respect to the race of the defendants have been raised. Therefore, it is of interest to develop novel methods to produce predictions while avoiding disparate treatment on the basis of the protected attribute, race. More precisely, we wish to ensure that the demographic parity constraint is satisfied, which, in this context, simply dictates that the weighted empirical CDFs of the race-specific fitted recidivism scores are *identical* or *similar*.

For simplicity of exposition, we only consider two levels for the protected attribute of race, namely African-American and non-African-American, and consider a sub-sample of the entire dataset with 100 defendants corresponding to each level of the protected attribute. As a covariate, for each defendant, we consider the demographic information—sex (binary), age (continuous), and marital status (categorical)—and criminal status—legal status (categorical), supervision level (categorical), and custody status (categorical). We use linear regression (i.e., *h* is linear in the covariates) as our predictive model of choice; the methodology readily extends to more complicated models. The raw recidivism scores for African-Americans versus non-African-Americans show a clear discrepancy). We shall assess the relative performance of the *in-model* scheme in (Equation 3) and the *two-step* scheme in (Equation 4) and (Equation 5) in ensuring demographic parity with respect to the protected attribute of race (refer to Figure 3 and Figure 4). When we fit the predictive model without any fairness constraint, the fitted empirical cumulative distribution functions corresponding to the two sub-populations are widely different. Our *in-model* scheme, as well as the *two-step* scheme, significantly reduces the discrepancy owing to the in-built fairness-based regularization. As expected, the *in-model* scheme provides slightly lower bias since it performs the two-step optimization simultaneously.

## 5. Entropy-Based Portfolio Allocation

We present an application of the proposed parametric distribution-guided entropy maximization framework to portfolio allocation problems [5,52,53]. Portfolio optimization is concerned with the allocation of an investor’s wealth over several assets to optimize specific objective(s) based on historical data on asset returns. To elucidate the problem clearly, let $R_{\left(i\right)}={\left(R_{i,1},R_{i,2},\ldots ,R_{i,d}\right)}^{\prime }$ be the excess returns on *d* risky assets recorded over time i∈[n]. The portfolio $\left(w_{1},\ldots ,w_{d}\right)$ is a vector of weights that represents the investor’s relative allocation of their wealth satisfying $\sum_{i=1}^{d}w_{i}=1$ and $w_{i}\geq 0,i\in \left[d\right]$. The goal is to learn the $\left(w_{1},\ldots ,w_{d}\right)$ subject to specific constraints based on historical data.

### 5.1. Related Works

Markowitz’s mean variance optimization [53] is widely recognized as one of the foundational formulations of the portfolio selection problem. The traditional mean variance (MV) optimal portfolio weights [53] are obtained via
$$\begin{array}{c} {\mathrm{argmax}}_{w}\left[w^{T}\mu -\frac{\lambda }{2}w^{T}\Sigma w\right], \end{array}$$
such that $\sum_{i=1}^{d}w_{i}=1$, where $\mu ={\left(\mu_{1},\ldots ,\mu_{d}\right)}^{T}=\left(1/n\right)\sum_{i=1}^{n}R_{\left(i\right)}$ and $\Sigma =\left(1/n\right)\sum_{i=1}^{n}\left(R_{\left(i\right)}-\mu \right)$${\left(R_{\left(i\right)}-\mu \right)}^{T}$ are the mean and variance of the historical return, and λ>0 is a risk aversion parameter. Given a specific mean and covariance matrix, the Markowitz paradigm offers an elegant approach to achieve an efficient allocation, where the pursuit of higher expected returns inevitably entails greater risk. However, in this framework, it is essential either for the asset returns to follow a normal distribution or for the utility to solely depend on the first two moments.

Real-world financial returns, as indicated by empirical evidence [64], diverge from normal distribution assumptions and commonly exhibit heavier tails and a lack of symmetry. For instance, Volatility Feedback Theory [36] posits that market volatility can influence subsequent returns. Specifically, it suggests that periods of high volatility can lead to skewed returns, where extreme price movements are more likely to occur. This theory underscores the idea that heightened volatility tends to be associated with increased risk and uncertainty in financial markets, potentially resulting in non-normal return distributions characterized by asymmetry and fat tails. Skewed returns indicate that the probability of extreme events, either positive and negative, is higher than what would be expected under a normal distribution. Consequently, skew normal distributions are routinely utilized to model observed returns [37,38].

Specifically, in the context of portfolio optimization, refs. [65,66] proposed to utilize higher-order moments in the portfolio allocation problem. However, portfolios created using sample moments of stock returns tend to be excessively concentrated in a small number of assets, which contradicts the fundamental principle of diversification. To that end, several approaches are proposed in the literature that ensure the shrinkage of the portfolio weights towards maximum diversification [5,67,68], i.e., they maximize the entropy of the portfolio weights. In particular, ref. [5] proposed to obtain the portfolio weights solving the optimization problem ${arg max}_{w}H_{d}\left(w\right)$ subject to $\sum_{i=1}^{d}w_{i}\mu_{i}\geq \mu_{0},\;w^{T}\Sigma w\leq \sigma_{0}^{2}$, such that $\sum_{i=1}^{d}w_{i}=1$ and $\left(\mu_{0},\sigma_{0}^{2}\right)$ are the target mean and variance of the portfolio return. In essence, this approach constitutes obtaining the portfolio weight via entropy maximization subject to moment-based constraints.

### 5.2. Proposed Methodology

Importantly, empirical evidence suggests that there is merit in modeling asset returns via non-normal distributions [65,69], e.g., a skew normal distribution [26,70]. However, it is often unfeasible to place more flexible constraints on the portfolio weights in terms of moment conditions. In this section, we intend to provide additional flexibility to the entropy-based portfolio optimization framework via providing the scope for statistical distance-based parametric distribution-guided constraints. Our semi-parametric framework provides a formidable alternative to the existing literature, since (a) we can flexibly specify the distribution of the expected return and (b) the entropy provides direct handling of portfolio diversity. We achieve this by obtaining portfolio weights via the optimization problem ${arg max}_{w}H_{d}\left(w\right)$ subject to $W_{2}^{2}\left[\frac{1}{n}\sum_{i=1}^{T}\delta_{w^{T}R_{\left(i\right)}}\left(\cdot \right),f_{\theta_{0}}\right]\leq \varepsilon$, such that $\sum_{i=1}^{d}w_{i}=1$. Here, $\frac{1}{n}\sum_{i=1}^{n}\delta_{w^{T}R_{\left(i\right)}}\left(\cdot \right)$ is the empirical distribution of the portfolio return, $f_{\theta }$ is the centering parametric family of the distribution of choice, $\theta_{0}$ is the fixed target value of θ, and ε is a user-defined parameter. For practical purposes, it is useful to express the optimization problem above as the following:(6)$$\begin{array}{cc} \underset{w}{arg min} & \left[\left(1-\lambda^{★}\right)W_{2}^{2}\left(\frac{1}{n}\sum_{i=1}^{n}\delta_{w^{T}R_{\left(i\right)}}\left(\cdot \right),\;f_{\theta_{0}}\right)-\lambda^{★}\;b_{d}H_{d}\left(w\right)\right]\; \end{array}$$
such that $\sum_{i=1}^{d}w_{i}=1$ and $b_{d}=1/\log d$. This choice of $b_{d}$ is convenient since it ensures that $b_{d}H_{d}\left(w\right)\in \left[0,1\right]$. Further, the user-defined parameter $\lambda^{★}\in \left[0,1\right]$ controls the balance between the portfolio diversity and the extent of deviation from the target distribution $f_{\theta_{0}}$.

For exposition in this article, we choose $f_{\theta_{0}}$ to be a skew normal distribution [26] with parameters $\theta ={\left(\omega ,\zeta ,\alpha \right)}^{\prime }$. The probability density function of a skew normal distribution SN(ζ,ω,α) is given by $f\left(z\right)=\frac{2}{\omega }\phi \left(\frac{z-\zeta }{\omega }\right)\Phi \left(\alpha \left(\frac{z-\zeta }{\omega }\right)\right),\;z\in R$ where ϕ(·) and Φ(·) are, respectively, the probability density function and cumulative density function of the standard normal distribution. For α=0, we can recover the normal distribution as the absolute value of skewness increases and the absolute value of α increases. For α>0, the distribution is left-skewed, and it is right-skewed for α<0. If Z∼SN(ζ,ω,α), then we have $\mu_{0}=E\left(Z\right),\sigma_{0}^{2}=Var\left(Z\right)$, $\gamma_{0}=\mathrm{Skewness}$. This allows us to set (ζ,ω,α) to achieve target $\theta_{0}=\left(\mu_{0},\sigma_{0}^{2},\gamma_{0}\right)$ of the portfolio return distribution. This resulting skew normal density with fully specified parameters then serves as the target distribution to calculate the portfolio weights based on (Equation 6). The user can select any flexible probability distribution to model the portfolio return and follow the prescribed procedure to compute the target parameter values. The suggested methodology could potentially be utilized with more versatile target distributions, such as the generalized skew normal distribution [71]. Nevertheless, addressing this aspect is beyond the scope of the current paper, although it is a promising avenue for future investigation.

### 5.3. Historical Stock Returns Data Analysis

We consider the stock returns data of 5 companies (AMZN, AAPL, XOM, T, MS) for the period of January 2000 to December 2020, publicly available from Yahoo! Finance. The data are aggregated at a monthly level. The goal is to compare the mean variance optimal portfolio and the proposed parametric distribution-guided portfolio allocation framework. First, we compute the mean variance optimal portfolio for varying values of the risk aversion parameter λ∈[0,10]. Figure 5 records the skewness, excess kurtosis, and number of zero portfolio weights for the mean variance optimal portfolio for varying λ. We focus on λ set at 1—a choice at which 3 out of 5 portfolio weights are 0, and the optimal portfolio return distribution is negatively skewed and leptokurtic. This exposes the fact that, once we have fixed the λ, the mean variance optimal portfolio optimization framework does not offer direct control over the portfolio diversity, and we will potentially obtain portfolio allocations concentrated on very few assets. Next, we fix the parameters of a skew normal density $\theta ={\left(\omega ,\zeta ,\alpha \right)}^{\prime }$ such that its mean, variance, and skewness match the quantities of the mean variance optimal portfolio return at λ=1. Finally, we compute the skew normal distribution-guided maximum entropy portfolio for varying values of the balance parameter $\lambda^{★}\in \left[0,1\right]$ in (Equation 6). Figure 6 presents the entropy of the portfolio weights and the departure of the portfolio return distribution from the guiding skew normal distribution as a function of $\lambda^{★}\in \left[0,1\right]$. This showcases that, contrary to the mean variance optimal portfolio allocation, here, the fund manager can choose a specific $\lambda^{★}$ to ensure the desired level of portfolio diversity, while maintaining fidelity towards a pre-specified distribution of the portfolio return distribution.

## 6. Concluding Remarks

We introduce a nonparametrically oriented framework that aims to align the maximum entropy weight-adjusted empirical distribution of observed data closely with a predefined and potentially continuous probability distribution, while permitting mild deviations. The framework’s versatility is showcased in three distinct applications. We anticipate the proposed methodology’s utility in numerous other statistical tasks requiring data re-weighting, e.g., robustness [18], covariate shifts [18], ill-posed inverse problems [4], etc.

## Figures and Tables

**Figure 1 entropy-26-00249-f001:**
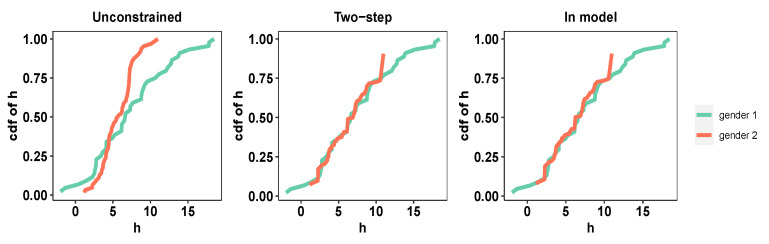
***Distress Analysis Interview Corpus.*** Empirical CDFs of fitted *h* for the two groups, with no fairness constraint $\left(W_{2}=19.32\right)$, fair post-processing $\left(W_{2}=2.24\right)$, and fair model fitting with $(W_{2}=0.79$), respectively, at $\lambda^{★}=0$.

**Figure 2 entropy-26-00249-f002:**
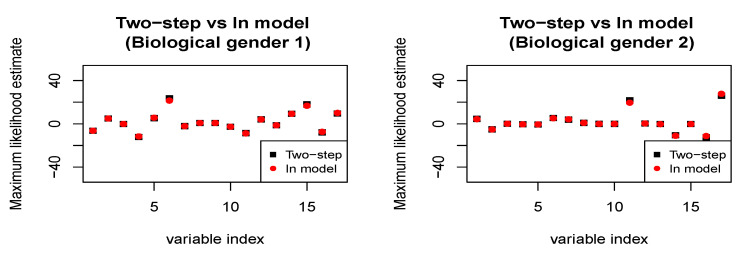
***Distress Analysis Interview Corpus.*** Maximum likelihood estimates of the regression coefficients under both two-step and in-model schemes. In the in-model scheme, the estimates are slightly modified since the regression coefficients and the weights assigned to the data are learned simultaneously. For details on the in-model and two-step approaches, refer to Equations (Equation 3), (Equation 4), and (Equation 5), respectively.

**Figure 3 entropy-26-00249-f003:**
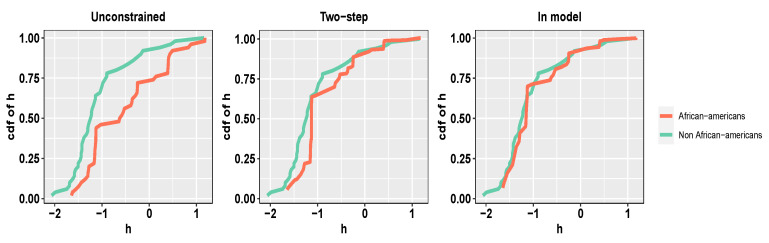
***COMPAS dataset.*** Empirical CDFs of fitted *h* for the two groups, with no fairness constraint $\left(W_{2}=0.72\right)$, fair post-processing $\left(W_{2}=0.05\right)$, and fair model fitting with $\left(W_{2}=0.02\right)$, respectively, at $\lambda^{★}=0$.

**Figure 4 entropy-26-00249-f004:**
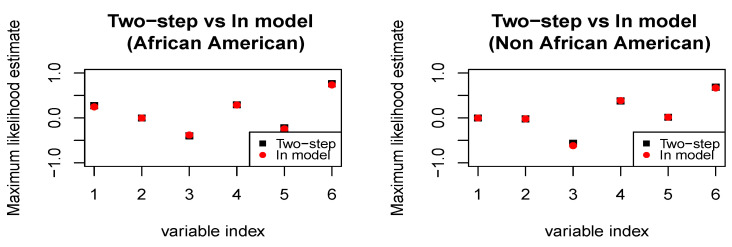
***COMPAS dataset.*** Maximum likelihood estimates of the regression coefficients under both two-step and in-model schemes. In the in-model scheme, the estimates are slightly modified since the regression coefficients and the weights assigned to the data are learned simultaneously.

**Figure 5 entropy-26-00249-f005:**
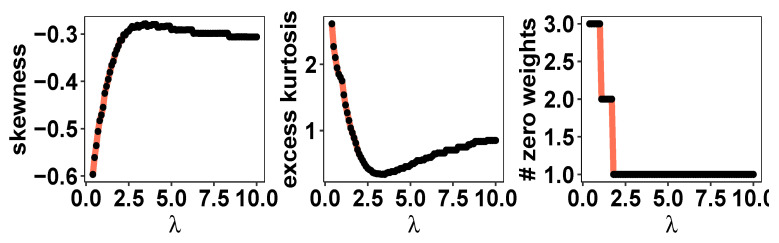
Limitations of mean variance optimal portfolio. (i) The skewness and excess kurtosis plots provide evidence that the normality assumption for expected returns does not hold. (ii) A small value of λ leads to zero weight to several assets.

**Figure 6 entropy-26-00249-f006:**
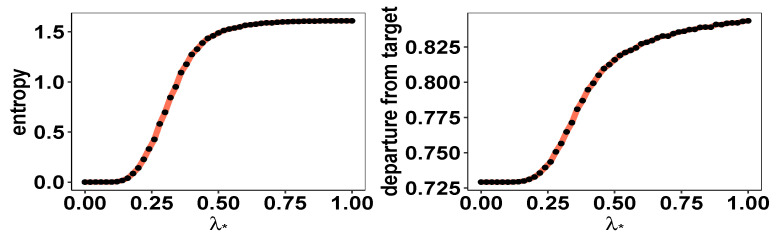
With a fixed target skew normal return, varying values of $\lambda^{★}\in \left[0,1\right]$ provide different balances between diversity and departure from the target. The desired degree of diversification can be achieved $\lambda_{★}$ via a simple grid search on $\lambda_{★}\in \left[0,1\right]$.

**Table 1 entropy-26-00249-t001:** Average bias (=$\left|\right|\left(\mu ,\sigma^{2}\right)-\left(\hat{\mu },\hat{\sigma^{2}}\right)\left|\right|$) and coverage (within brackets) of the MLE, PMLE, BPPE [56], and BDCM estimators for varying data generating mechanisms.

*n*	ρ	0.1	0.5	0.8
500	MLE	0.19 (0.91)	0.68 (0.48)	1.11 (0.45)
	BPPE	0.67 (0.82)	0.65 (0.72)	0.71 (0.78)
	PMLE	0.16 (0.94)	0.16 (0.91)	0.16 (0.94)
	BDCM	0.16 (0.92)	0.16 (0.93)	0.16 (0.95)
1000	MLE	0.16(0.87)	0.69(0.48)	1.11(0.42)
	BPPE	0.15 (0.92)	0.18 (0.90)	0.18(0.92)
	PMLE	0.11 (0.94)	0.10 (0.94)	0.10 (0.92)
	BDCM	0.11 (0.93)	0.10 (0.94)	0.10 (0.96)
1500	MLE	0.15(0.84)	0.68(0.47)	1.11(0.42)
	BPPE	0.12 (0.94)	0.10 (0.89)	0.12 (0.90)
	PMLE	0.09(0.94)	0.08(0.94)	0.07 (0.93)
	BDCM	0.09(0.94)	0.08(0.93)	0.08(0.94)
2000	MLE	0.15 (0.81)	0.68 (0.48)	1.10 (0.40)
	BPPE	0.09 (0.92)	0.08 (0.92)	0.07(0.92)
	PMLE	0.07(0.95)	0.07(0.95)	0.06(0.92)
	BDCM	0.07 (0.95)	0.07 (0.97)	0.07 (0.97)
2500	MLE	0.15 (0.75)	0.68 (0.47)	1.10 (0.39)
	BPPE	0.06(0.94)	0.07(0.88)	0.06 (0.92)
	PMLE	0.06(0.96)	0.07 (0.94)	0.06 (0.92)
	BDCM	0.06(0.97)	0.06 (0.95)	0.06 (0.94)

**Table 2 entropy-26-00249-t002:** **NHANES data.** Bias $=||\beta -\hat{\beta }\left|\right|$ and coverage of the moment condition model-based estimates of the regression parameters for varying sample sizes.

*n*	250	500	1000	2000
Coverage	0.95	0.95	0.96	0.96
Bias	0.42	0.27	0.18	0.13

## Data Availability

Data are contained within the article.

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
