# Peer review of "Constrained Reweighting of Distributions: An Optimal Transport Approach"

_entropy, 2024, doi:10.3390/e26030249_

Round 1

Reviewer 1 Report

Comments and Suggestions for Authors

In this article, the authors present an innovative approach to addressing the issue by deploying a framework based on constrained entropy maximization, guided by a probability distribution. This method is flexible and improves the clarity of the results derived from the data analysis. The core idea is to adjust the empirical distribution of the observed data, through weighting, to closely match a specific family of probability distributions. This alignment is quantified using a measure of statistical distance.

The paper is well-written, and I have no reservations; therefore, I recommend acceptance

Reviewer 2 Report

Comments and Suggestions for Authors

Reviewer’s report on manuscript Entropy-2849206:

“Constrained Reweighting of Distributions:

an Optimal Transport Approach”,

by Chakraborty B., Bhattacharya A. and Pati D.

The paper uses an optimal transport approach to reweight the observed data and to alleviate for lack of randomness. The proposed method is applied to portfolio allocation, semi-parametric inference for complex surveys, and algorithmic fairness in machine learning algorithms.

Skew-normal distribution. Azzalini and Dalla Valle (1996) introduced the multivariate skew-normal distribution, but in the present paper the univariate version is used, which dates back to Roberts (1966). I wonder whether the proposed approach might be used with more general distributions, as for example the generalized skew-normal distribution (Loperfido, 2004). I reckon that the answer to this question falls outside the scope of the present paper, but the authors should at least consider it as a possible direction of future research.

Financial econometrics. According to the Volatility Feedback Theory (Brown et al, 1988) stock price reactions to unfavorable events tend to be larger than reaction to favorable events. This determines asymmetry which can be modelled by the skew-normal distribution (De Luca and Loperfido, 2004). This situation falls into the realm of nonrandom samples addressed by the present paper. For example, the COVID pandemic and the 2007/2008 financial crisis led to less-than-representative financial data.

Additional references

Brown, K. C., Harlow, W. V. and Tinic, S. M. (1988). Risk aversion, uncertain information, and market efficiency. Journal of Financial Economics, 22, 355-385.

De Luca, G. and Loperfido, N. (2004). A Skew-in-Mean GARCH Model for Financial Returns. In “Skew-Elliptical Distributions and Their Applications: A Journey Beyond Normality”, CRC/Chapman & Hall, 205-222.

Loperfido, N. (2004). Generalized Skew-Normal Distributions. In “Skew-Elliptical Distributions and Their Applications: A Journey Beyond Normality”, CRC/Chapman & Hall, 65-80.

Roberts C. (1966). A correlation model useful in the study of twins. Journal of the American Statistical Association 61, 1184-1190.

Reviewer 3 Report

Comments and Suggestions for Authors

Title: Constrained Reweighting of Distributions: an Optimal Transport Approach

By: Abhisek Chakraborty, Anirban Bhattacharya, Debdeep Pati

Submitted to: Entropy,   Ms I.d. Entropy-2849206

Report     02/10/2024

The authors introduce a nonparametrically imbued distributional constraints on the 

weights of empirical distribution, and develope a general framework leveraging the 

maximum entropy principle and tools from optimal transport, so that the maximum entropy 

weight adjusted empirical distribution of the observed data is close to a pre-specified 

probability distribution in terms of the optimal transport metric while allowing for 

subtle departures.

Major Comments:

* The work is closely related to empirical likelihood, a very popular nonparametric

  approach of modeling, inference, in various settings. It covers a broad range of 

  statistical questions. See Owen (1988, 1990, 1991, 2001) and others (see the reference

  list at the end). In fact, the work can viewed as a variation of extenition of the 

  empirical likelihood. I'm surprised that the authors did not mention works in this 

  field.

* Given the above comments, the authors should compare their work with that of 

  empirical likelihood, discuss their similarities and discrepencies; the advantage(s)

  and dis-advantage(s) of each method.

* Preferrablly, the authors are encouraged to derive the asymptotic properties of the

  proposed method, and compare with those from the empirical likelihood.

References

Chen, J. and Qin., J. 1993, Empirical likelihood estimation for finite populations and

   the effective usage of auxiliary information. Biometrika 80, 107-116.

Chen, S.X. 1997, Empirical likelihood for nonparametric density estimation. Australian 

   Journal of Statistics 39, 47-56.

Imbens, G. 2002, Generalized method of moments and empirical likelihood. Journal of 

   Busyness and Economic Statistics 20(4), 493-506.

Kitamura, Y. 2001, Asymptotic optimality of empirical likelihood for testing moment 

   restrictions.  Econometrica 68, 1661-1672.

Kolaczyk, E.D. 1994, Empirical likelihood for generalized linear models. Statistica 

   Sinica 4, 199-218.

Lazar, N.A. and Mykland, P. 1999, Empirical likelihood in the presence of nuisance 

   parameters.  Biometrika 86 (1), 203-211.

Owen, A.B. 1988, Empirical likelihood ratio confidence intervals for a single functional. 

   Biometrika 75, 237-249.

Owen, A.B. 1990, Empirical likelihood confidence regions. Annals of Statistics 18, 90-120.

Owen, A.B. 1991, Empirical likelihood for linear models. Annals of Statistics 19, 1725-1747.

Owen, A.B. 2001, Empirical Likelihood, Chapman & Hall.

Qin, G.S. and Tsao, M. 2005, Empirical likelihood based inference for the derivative of 

   the nonparametric regression function. Bernoulli 11, 715-735.

Qin, J. 1993, Empirical likelihood in biased sample problems. Annals of Statistics 21, 

    1182-1196.

Qin, J. and Lawless, J.L. 1994, Empirical likelihood and general estimating equations. 

    Annals of Statistics 22, 300-325.

Comments on the Quality of English Language

see my report

Round 2

Reviewer 2 Report

Comments and Suggestions for Authors

You satisfactorily addressed all my remarks.

Author Response

Thanks!

Reviewer 3 Report

Comments and Suggestions for Authors

Title: Constrained Reweighting of Distributions: an Optimal Transport Approach

By: Abhisek Chakraborty, Anirban Bhattacharya, Debdeep Pati

Submitted to: Entropy,   Ms I.d. Entropy-2849206-v2

Report     03/13/2024

In this revision, the authors addressed most of my comments, discussed the relationship

of their method with the empirical likelihood, and the advantege(s) and dis-advantage(s)

of each method. It is still desirable to mention this relationship and advantage(s) in the Abstact.

Comments on the Quality of English Language

see my review

Author Response

Thanks for your suggestions! We have added a short discussion on the relationship of the proposed methodology with the EL literature. We have also delineated the additional flexibilities that our proposal provides.